# Antioxidant Potential of the Sweet Whey-Based Beverage *Colada* after the Digestive Process and Relationships with the Lipid and Protein Fractions

**DOI:** 10.3390/antiox11091827

**Published:** 2022-09-16

**Authors:** Victoria E. García-Casas, Isabel Seiquer, Zaira Pardo, Ana Haro, Isidra Recio, Raquel Olías

**Affiliations:** 1Facultad de Ingeniería Química, Universidad de Guayaquil, Ciudadela Universitaria, Avenida Delta S/N y Avenida Kennedy, Guayaquil 090514, Ecuador; 2Departamento Nutrición y Producción Animal Sostenible, Estación Experimental del Zaidín, Consejo Superior de Investigaciones Científicas (CSIC), San Miguel 101, Armilla, 18100 Granada, Spain; 3Instituto de Investigación en Ciencias de la Alimentación, CIAL (CSIC-UAM, CEI UAM+CSIC), Nicolás Cabrera, 9, 28049 Madrid, Spain

**Keywords:** sweet whey, whey beverages, antioxidant activity, in vitro digestion, fatty acids, protein profile, antioxidant peptides

## Abstract

Whey-based beverages could be an effective way of reusing a by-product of th cheese industry, mitigating environmental hazards and, at the same time, profiting a useful food with high nutritional and antioxidant properties. In this study, a traditional Ecuadorian beverage (*Colada*) was prepared combining sweet whey, Maracuyá and barley. Antioxidant properties before and after an in vitro digestion using the INFOGEST method were determined, and relationships with intestinal transformations of the lipid and protein fractions were analyzed. The digestive process had a positive effect on antioxidant properties based on increased values of ABTS and FRAP located in the bioaccessible fraction (BF), together with strong increments of total polyphenols. Moreover, pretreatment of Caco-2 cells with the BF of *Colada* significantly reduced ROS generation (*p* < 0.001) measured by the dichlorofluorescein assay. Substantial changes of the fatty acid profile occurred during digestion, such as a fall of saturated fatty acids and a rise of polyunsaturated. The protein profile, examined by SDS-PAGE and exclusion molecular chromatography in the BF, showed that the major part of the proteins were digested in the intestinal phase. Analysis of NanoLC-MS/MS revealed 18 antioxidant peptides originated from whey proteins, but also 16 peptides from barley with potential antioxidant properties. In conclusion, combining sweet whey with Maracuyá and barley constitutes an excellent nutritional beverage with a strong antioxidant potential.

## 1. Introduction

Whey is the remaining liquid of the milk after the separation of curd during cheese manufacturing, accounting for 85–90% of the milk volume [1]. It is the main byproduct of the cheese-making industry and about 50% of it is treated as a waste and discarded as an effluent, causing major environmental pollution problems [2]. However, whey contains approximately 55% of the nutrients from milk, including 50–100% of lactose (depending on the pH of whey), 20% of the total protein content and minerals, and it is recognized as a source of functional and bioactive compounds which may represent a valuable food ingredient for human health [3]. Recently, great attention has been focused on the benefits of whey protein regarding antioxidant properties [2,4,5,6]. Dietary antioxidants provide health advantages to the consumer by combating oxidative stress through different pathways and thus reducing the risk of related diseases, such as cardiovascular and inflammatory pathologies [7].

The whey protein fraction, composed by β-lactoglobulin, α-lactalbumin, bovine serum albumin, immunoglobulins and lactoferrin, is considered to be the main factor responsible for the antioxidant activity [8]. Antioxidant properties of bovine whey proteins have been extensively proven in vitro by decreasing lipid peroxidation [9] or scavenging free radicals [10,11] and have been attributed to their hydrophobic and aromatic amino acids which can serve as proton donators for free radicals [12]. Assays in cell cultures have shown that whey proteins protect against oxidation by stimulating the cellular antioxidant defenses, such as glutathione peroxidase, catalase and superoxide dismutase activities, and enhance expression of genes linked to the antioxidant capacity of cells such as HO-1, NQO1 and Nrf2 [13,14].

The current scientific evidence suggests that whey has transitioned from a waste into a high value-added product useful for the food industry [1]. Nutritional properties of whey have aroused worldwide interest in the formulation of whey-based beverages enriched with diverse fruits and cereals [15]. A wide number of whey beverages have been developed in recent years, those containing fruit juices being among the more relevant and are often preferred by consumers [16,17]. Moreover, fruits and vegetables are a valuable source of different antioxidant compounds, such as carotenoids and polyphenols [17]. *Passiflora edulis* var. Flavicarpa, also known as yellow passion fruit or Maracuyá, is a highly desirable fruit due to the eating quality of its fruits, juiciness, attractive nutritional values and essential health benefits, including antioxidant activity [18]. Among cereals, barley (*Hordeum vulgare L*.) is characterized by its high content of saponarin (flavone-C-glycosides) which possesses antioxidant effects, policosanol polyphenol series, minerals and free amino acids [19]. In developing countries, such as Ecuador, whey is often discarded without any treatment because of the high cost of processing it, with the consequent adverse environmental impacts and wasting of a valuable product for a population with serious problems of maternal and child malnutrition [20]. *Colada* is a traditional beverage usually consumed in Ecuador, made from fruit juice, cereals and spices (https://soleq.travel/traditional-ecuadorian-drinks-desserts/, accessed on 1 August 2022).

It is important to bear in mind that, once ingested, the bioactive compounds of any food and their metabolites are subjected to transformations in the gastrointestinal tract due to enzymatic, mechanic and acidic conditions, which may alter their intrinsic activity [21]. It is widely accepted that the crucial requisite for a bioactive compound to exert an antioxidant activity in vivo is to be bioaccessible, i.e., released from the food matrix during the digestive process and still maintain its activity [21]. In this sense, it has been shown that the antioxidant capacity of whey isolated proteins significantly increases after in vitro gastrointestinal digestion and that α-lactalbumin, which releases the highest amount of tryptophan, has the highest antioxidant properties [5].

In the present work, we considered that using whey as a substitute for water in the *Colada* drink would have a double advantage. On the one hand, it would profit a by-product of the cheese industry, mitigating environmental hazards and favoring a circular economy in a sustainable way and, on the other hand, it would increase the nutritional value of the traditional beverage, producing a novel valuable product with nutritional benefits for population. With this background, the aim of this study was to determine the antioxidant properties of a beverage made from sweet whey and enriched with Maracuyá (*Passiflora edulis*) and barley grain (*Hordeum vulgare*), before and after in vitro digestion. Within this purpose, modifications of the lipid and protein fractions due to the digestive process were also analyzed.

## 2. Materials and Methods

### 2.1. Chemicals

The chemicals used were all of high purity or analytical reagent grade. Bidistilled deionized water (Milli-Q purification system, Millipore, Bedford, MA, USA) was used. Methanol was provided by VWR (Barcelona, Spain), while sodium bicarbonate, acetate sodium, galic acid and hydrochloric acid were acquired from Merck (Darmstad, Germany). Folin–Ciocalteau reagent, 6-hydroxy-2,5,7,8-tetramethyl-chroman-2-carboxylic acid (Trolox), 2,2-diphenyl-1-picrylhydrazyl (DPPH), amylase (A1031), pepsin (P7012), pancreatin (P7545), bile salts, sodium dodecyl sulfate (SDS), o-phthaldialdehyde (OPA), DL-dithiothreitol (DTT), L-serine, (4-(2-hydroxyethyl)-1-piperazineethanesulfonic acid) (HEPES), tert-butylhydroperoxide (*t*-BOOH), the cell culture media, cell culture-grade chemicals and fatty acid standards (C15:0 and CRM47885 Supelco 37 Component FAME Mix) were purchased from Sigma (Sigma-Aldrich, St Louis, MO, USA). The ABTS was obtained from Amresco (Solon, OH, USA). 2,4,6-Tri(2-pyridyl)-s-triazine (TPTZ) and iron (III) chloride for the ferric reducing power (FRAP) assay were from Fluka Chemicals (Fluka Chemicals, Madrid, Spain).

### 2.2. Preparation of the Whey-Based Beverage Colada and Sensory Evaluation

Milk was obtained from braham cows and fresh bovine whey was supplied by the DIPROLAC center (Balzar canton, Guayas Province, Ecuador). The sweet whey obtained after the precipitation of casein by the enzymatic action of rennet was collected at 37 °C and pasteurized at 65 °C for 30 min, cooled with thermal shock and kept refrigerated at 4 °C until used. The pH (6.55) was measured with a pH-meter BioCharge (BioCharge C.A., Maracay, Venezuela) and it was within the limits of sweet whey (6.4–6.8) of the Ecuadorian technical standard NTE INEN 2594. Maracuyá, barley and spices used in the beverage preparation were purchased in local stores (municipal market “San Jacinto”, Guayaquil, Ecuador); all of them of Ecuadorian origin and with food quality. Barley was selected to be free of contaminants such as weevils, molds or stones, and free of moisture; Maracuyá fruits were collected at a coloration degree of 3–4, according with the maturity degree.

The *Colada* was prepared following the traditional protocol for drinks with cereals and fruits used regularly in the Ecuadorian gastronomic culture, with the exception that water was replaced by sweet whey. Since unprocessed whey is not attractive to consumers because of its sensory properties [22], previous sensorial evaluation was performed to determine the consumer acceptance of the whey beverage. Initially, formulations with different proportions of Maracuyá juice (5, 7, 10, 15 and 20%) were evaluated by five trained judges applying discriminative sensorial analysis to choose the best formulation (Appendix A). The juice was obtained after separating the seed from the pulp with the help of a spoon and passing the pulp through a 0.5 mm mesh size sieve. The beverage with 15% Maracuyá juice obtained the highest score and, therefore, subsequent sensory analysis of this formulation was performed with a panel of 80 untrained consumers. The sensory attributes evaluated were flavor (bitter, salty, acid, sweet, umami, pungent and astringent), color (brightness, transparency, opacity and paleness), texture (pasty, tender, rubbery, fluid, dense, viscous, elastic, sticky, adherent, fine and liquid) and general appearance (slight, dense and very dense). A 5-point hedonic scale was used in the following sequence: (1): It is not perceived, (2): It is perceived little, (3): It is perceived as fair, (4): It is perceived as good and (5): It is perceived as excellent [22,23]. Panelists were also asked whether they would consume the product, answering with yes or no. Once the good overall acceptance of the assayed whey-beverage *Colada* was proven (Appendix A)*,* this formulation was selected to perform the current study.

The ingredients (depicted in Table 1) were mixed and homogenized. The mixture was thermally treated at 87 °C for 15 min with constant stirring (Bunsen, Madrid, Spain), cooled to room temperature, filtered and lyophilized (freeze dryer Virtis Genesis, SQ25EL, Warminster, PA, USA). Then, the sample was reduced to a fine dried powder (Moulinex, Barcelona, Spain), mixed to obtain a homogenate sample and stored in aliquots at −20 °C until further analysis.

### 2.3. Analysis of Nutrient Composition

Analyses were performed in quadruplicate. Dry matter (method 934.01) and total ash content (method 942.05) were determined using official methods (AOAC, 2000). Fat was extracted with chloroform:methanol at a ratio of 2:1 and quantified by Soxhlet (AOAC, 2000). Total nitrogen was analyzed by the Dumas procedure using LECO Truspec CN equipment (LECO Corporation, St. Joseph, MI, USA); protein content was calculated using the factor of 6.38. Gross energy was determined in an isoperibolic bomb calorimeter (Parr Instrument Co., Moline, LL, USA). The mineral content was analyzed according to the procedure described by Haro et al. [24]. Briefly, after wet digestion of the samples (HNO_3_:HClO_4_ 1:4, 180–220 °C), calcium (Ca), magnesium (Mg), sodium (Na) and potassium (K) were analyzed by flame atomic absorption spectroscopy (FAAS) in a Perkin–Elmer Analyst 700 Spectrophotometer (Norwalk, CT, USA). Total phosphorus (P) was determined colorimetrically at 820 nm in a spectrophotometer (Shimadzu UV-1700, Model TCC-240A, Columbia, CA, USA) by the vanadomolybdate procedure (AOAC, 2000). Certified external standards (European Commission, Reference Materials Unit, Geel, Belgium) were used to assess the accuracy of the methods. All glassware and polyethylene sample bottles used for mineral analysis were washed with 10 mM nitric acid and Milli-Q water.

### 2.4. In Vitro Digestion

The simulated digestion was performed in triplicate according to the established INFOGEST method [25,26]. Before digestion, an adequate amount of the lyophilized sample was resuspended in water to obtain a final weight of 5 g of fresh matter. The process consisted of oral, gastric and intestinal phases.

For the oral phase, 5 g of the beverage was mixed with 5 g of salivary fluid with α-amylase (75 U/mL). The sample was incubated at 37 °C for 2 min with mild and constant agitation. The gastric phase was initiated with the addition of simulated gastric fluid with pepsin (2000 U/mL in the final mixture); the pH was lowered to 3.0 and the final volume was adjusted to 20 mL with Milli Q water. The mix was then incubated at 37 °C for 2 h with mild and constant agitation. For the intestinal phase, the pH was raised to 7.0 and simulated intestinal fluid containing pancreatin (200 U/mL) and bile salts (10 mM) was added. The volume of the sample was brought up to 40 mL and the mixture was again incubated for 2 h at 37 °C with continuous shaking.

To stop the intestinal digestion, samples were immediately frozen in liquid nitrogen. Samples were defrosted afterwards on ice and centrifuged at 12,000× *g* at 4 °C to separate the soluble or bioaccessible fraction (BF) and the residual fraction (RF). A blank digestion was performed by using a mixture of simulated digestion fluids and the same concentration of enzymes, but the sample was replaced with Milli-Q water. Aliquots for antioxidant analysis and cell culture assays were stored at −80 °C in tubes protected from the light under a nitrogen blanket. For cell experiments and peptide analysis, the BF was filtered through a 30 KDa cutoff filter (Amicon ultra-15 15mL, millipore) in order to eliminate all the intestinal enzymes background.

### 2.5. Total Phenolic Content and Antioxidant Activity

Prior to analysis of antioxidant activity and total phenolic content (TPC), a chemical extraction was performed in the undigested samples and the RF from the simulated digestion, following the procedure described by Seiquer et al. [27]. 250 mg of the sample, or the total RF, were mixed with 2.5 mL of acidic methanol/water (50:50 *v*/*v*, pH 2), shaken for 60 min at 220 rpm (circulating shaker OVAN, Barcelona, Spain) and centrifuged (2500× *g*, 10 min, 4 °C). A second extraction was performed, adding to the residue 2.5 mL of acetone/water (70:30, *v*/*v*). The supernatants were combined and used for determinations. The BF was used directly.

Three procedures were applied to test the antioxidant activity of the samples: the ABTS and DPPH assays, to measure the free radical scavenger ability, and the FRAP method, to study the ferric reducing antioxidant power. The methodologies previously described by Seiquer et al. [27] were generally followed. A multilabel plate reader (Victor X3, Waltham, MA, USA) was used. Aqueous solutions of Trolox were used for calibration (0.01–0.1 mg/mL). The results were expressed as mM equivalents of Trolox per kg of sample.

In the ABTS assay, the ABTS•+ solution was prepared 12–16 h before use by mixing 2.45 mM potassium persulfate with ABTS 7 mM and diluted to an absorbance of 0.70 ± 0.02 at 750 nm with 5 mM phosphate-buffered saline. Twenty µL of the samples were added to 280 µL of ABTS solution and incubated for 20 min in the dark before reading the absorbance at 750 nm.

For the DPPH method, 50 µL of the sample were mixed with 250 µL of DPPH solution (74 mg/L in methanol prepared freshly). After 60 min of incubation period, the absorbance was read at 520 nm, maintaining the temperature at 30 °C. The FRAP reagent was prepared daily by mixing 10 mM Fe^2+-^2,4,6-Tri(2-pyridyl)-1,3,5-triazine (TPTZ) with 40 mM HCl, 20 mM ferric chloride and 0.3 M acetate sodium buffer (pH 3.6) at a ratio 1:1:10 *v/v/v*. Twenty µL of sample extract were added to 280 µL of warmed FRAP reagent (37 °C) and incubated at 37 °C for 30 min in darkness, and the absorbance was read at 595 nm.

The TPC was determined following the Folin–Ciocalteau colorimetric method [27]. Ten µL of sample and 10 µL of Folin–Ciocalteau reagent were mixed and let stand for 3 min. 200 µL of sodium carbonate solution (75 g/L) were added, the volume was made up to 250 µL with Milli-Q water, mixed and allowed to stand in the dark for 60 min. The absorbance was measured at 750 nm against a standard curve of gallic acid (0–250 mg/L).

### 2.6. Reactive Oxygen Species (ROS) Generation in Caco-2 Cells

The BF obtained after in vitro digestion of the *Colada* beverage was used to test the antioxidant potential of the whey beverage at the cell level by measuring effects in the reactive oxygen species (ROS) generation in Caco-2 cells, according with Borges et al. [21]. 

Caco-2 cells were purchased through the Cell Bank of Granada University (Granada, Spain) from the European Collection of Cell Cultures (ECACC). The cells were maintained in 75 cm^2^ plastic flasks (Costar, Cambridge, MA, USA) by serial passages, using as a culture medium high-glucose Dulbecco’s modified minimal essential medium (DMEM), with heat-inactivated fetal bovine serum (FBS) (10%), NaHCO_3_ (3.7 g/L), nonessential amino acids (1%), HEPES (15 mM), bovine insulin (0.1 UI/mL) and 1% antibiotic–antimycotic solution. ROS determinations were carried out both at basal conditions and against an induced oxidative stress. Experiments were carried out with BF:FSB-free DMEM (at a ratio of 1:1 *v/v*), as previous assays using the colorimetric MTT method (3-(4,5-dime thylthiazol-2-yl)-2.5-diphenyltetrazolium bromide, Roche, Mannheim, Germany) showed that cell viability was >90% at such conditions.

Determination of ROS generation was performed by the dichlorofluorescin (DCFH) assay. Briefly, cells were seeded in 24-well multiwell plates 10 × 10^4^ cells/mL (400 µL/well) and allowed to grow for 48 h. Spent medium was removed and cells were preincubated with the BF for 2 h. Then, cells were treated with DCFH 20 µM and incubated for 1 h. The DCFH was removed and culture medium (for basal measurements) or *t-BOOH* 10 mM (to induce oxidation) was added to the wells. The absorbance was measured at a wavelength of 485 nm excitation and 535 nm emission at 37 °C for 0–90 min. In the presence of free radicals such as ROS, the DCFH is oxidized into dichlorofluorescein (DCF) and emits fluorescence, which is measured to estimate the ROS production. Results of ROS generation were expressed as fluorescence units.

### 2.7. Fatty Acid (FA) Analysis

The FA analysis was carried out before and after the in vitro digestion of the samples. Prior to the analysis, aliquots of 10 mL of the BF (in triplicate) were lyophilized using a LyoQuest-85 freeze dryer (Telstar, Terrasa, Spain).

Fat was extracted with a 2: 1 chloroform/methanol mixture (*v*/*v*) [28], and the FA were then methylated in accordance with Kramer et al. [29], with slight modifications. The extracted lipid fraction was initially methylated using NaOH/methanol (50 °C, 15 min) followed by HCl/methanol (50 °C, 1 h) for obtaining the fatty acid methyl esters (FAME). Pentadecanoic acid (C15:0) was used as internal standard. The fatty acid profile was determined with a gas chromatograph (Focus GC, Thermo Scientific, Milan, Italy) equipped with a split/splitless injector, a flame ionization detector and a 100 m × 0.25 mm × 0.2 µm capillary silica gel column (SP 2560 Supelco, Bellefonte, PA, USA). The temperature of the program was 70 to 240 °C and the injector and detector were maintained at 250 °C. Helium was used as carrier gas at a flow rate of 1 mL/min. Individual FAME peaks were identified by comparing their retention times with those of standards (Supelco 37 Component FAME Mix) and the results were expressed as the relative percentage of each fatty acid of the total identified, calculated by internal normalization of the chromatographic peak area. Chromatograms of the FAME of the whey-beverage (*Colada*) before and after the in vitro digestion are shown as Appendix A, respectively).

### 2.8. Amino Acid Analysis

The amino acid (aa) analysis was carried out quantitatively by using a Biochrom 30 amino acid analyzer based on ion-exchange liquid chromatography and a post-column continuous reaction with ninhydrin. The post-column ninhydrin derivative eluted from the column was monitored at 570 and 440 nm (proline). The resultant chromatogram gave us the identity and amount of the aa present in samples (Appendix A). Total aa content was analyzed after 50 mg were hydrolyzed with 4 mL of 6 N HCl. The solutions were sealed in tubes under nitrogen and incubated at 110 °C for 24 h. The sulfur-containing AA, cysteine, and methionine were measured as methionine sulphone and cysteic acid after performic acid oxidation (Appendix A). The determination of tryptophan was not possible due to its degradation following acid hydrolysis.

### 2.9. Degree of Hydrolysis (DH)

The protein hydrolysis in each one of the digestion phases was measured using the OPA (o-phthaldialdehyde) methodology at 340 nm [30]. The OPA reagent was prepared as follows: 7.62 g di-Na-tetraborate decahydrate and 200 mg Na-dodecyl-sulfate (SDS) were dissolved in 150 mL deionized water. 160 mg OPA was dissolved in 4 mL ethanol; the OPA solution was then transferred to the above-mentioned solution. Finally, 176 mg DTT was added to the solution. The solution was made up of 200 mL with deionized water. The free amino group concentration was determined with reference to a calibration curve using l-serine (12.5–100 mg L^−1^), since in OPA reactions serine shows a response close to the average response of amino acids. The degree of protein hydrolysis (DH%) was estimated according to the following equation:DH % = [NH_2_ (final) – NH_2_ (initial)] × 100 / NH_2_ (acid) – NH_2_ (initial)]
where NH_2_ (final) is the concentration of free amino groups in the digested sample after each phase, NH_2_ (initial) is the concentration of free amino groups before digestion, and NH_2_ (acid) is the total content of completely hydrolyzed sample in 6 N HCl at 110 °C for 24 h. All measurements were carried out at least in triplicate for each digesta.

### 2.10. SDS-PAGE Analysis of Protein Profiles

Denaturing gel analyses were carried out using gradient 4–12% Bis-Tris pre-cast gels (Invitrogen), according to the manufacturer’s instructions, with 2-N-morpholine-ethane sulphonic acid (NuPAGE MES, Invitrogen, Barcelona, Spain) as running buffer. Gels were stained using Colloidal Blue (Expedeon, Harston, UK). For accurate molecular weight estimation, the unstained protein standard Mark12TM (Invitrogen, LC5677, Barcelona, Spain) was included in the gel with proteins in the range of 2.5 to 200 kDa.

### 2.11. Gel Filtration Analysis (SEC)

Molecular weight of the soluble fractions after each digestion phase was monitored by size exclusion chromatography (SEC). Samples were loaded onto a HiPrep 26/60 Sephacryl S-100 HR column (flow rate of 0.3 mL per min) in 50 mM phosphate buffer containing 150 mM NaCl, pH 7.5. Three replicates per sample were analyzed. The column was calibrated by molecular weight standards, each in 2 mL buffer [bovine serum albumin (63.5 kDa) ovalbumin (48.1 kDa), chymotrypsinogen A (20.4 kDa), ribonuclease A (15.6 kDa), aprotinin (6.5 kDa), conalbumin (1.3 KDa).

### 2.12. Antioxidant Peptides Identification

Peptides after in vitro digestion were identified in the BF using liquid chromatography and mass spectrometry (nanoLC-MS/MS). The analysis was performed on a nanoLC (easy nanoLC II, Proxeon, now Thermo Scientific, West Palm Beach, FL, USA), directly connected to an Ion Trap Mass Spectrometer (Amazon Speed ETD, Bruker, Madrid, Spain), with a CaptiveSpray Source. Peptides separation was performed on a C18 column 75 µm × 15 cm, 3 µm, 100A (Acclaim PepMap100, Thermo Scientific, West Palm Beach, FL, USA) over a 180 min acetonitrile gradient from 5 to 30% B (solvent A: 0.1% formic acid in water; solvent B: 0.1% formic acid in acetonitrile) with a flow rate of 300 nl/min. The Ion Trap was set to analyze the survey scans in the mass range m/z 250 to 2500 in Enhanced Resolution MS mode (speed 8100 m/z/s), and the top 10 most intense ions in each duty cycle were selected for MS/MS in UltraScan mode (32,500 m/z/s). Fragmentation parameters: Active exclusion after 2 spectra and active release of 0.4 min, scan range: 50 m/z–2 × precursor.

Identification of the parental proteins for peptide sequences was done using the software ProteinScape (Bruker) and MASCOT v 2.7 (Matrix Science) search engine. Only peptide sequences with high confidence (>95%) were analyzed. Peptide sequences obtained were analyzed from BIOPEP-UWM, an online database of bioactive peptides (http://www.uwm.edu.pl/biochemia/index.php/pl/biopep, accessed on 10 July 2022). The BIOPEP-UWM database of bioactive peptides contained a current number of 4485 bioactive peptides at the time of the sequence-based search [31]. Peptides not reported in the BIOPEP data base were evaluated by Peptide Ranker, an online tool that allows the user to predict the probability of a peptide having a biological activity (Bioware.ucd.ie). Theoretical bioactivity was expressed as score values calculated from 0 to 1, with 1 being the most likely to be bioactive. Peptides with a score above 0.5 are considered to have a potential biological activity and were analyzed for antioxidant activity using the online tool AnOxPepre-1.0 (https://services.healthtech.dtu.dk/service.php?AnOxPePred-1.0, accessed on 20 July 2022), including free radical scavenging (FRS) and metal-chelating activity (MCA) scores, with its predictions being from 0 (not antioxidant) to 1 (antioxidant) [32].

### 2.13. Statistical Analysis

The experimental results are expressed as the mean ± standard error (SE) of at least three replicates. The data obtained were analyzed by applying analysis of variance (one-way ANOVA) to study the effects of the in vitro digestion. LSD test was used to compare mean values and significant differences were established at *p* < 0.05. Statistical calculations were carried out using the StatGraphics Centurion XVI software version 16.1.18 (StatPoint Technologies Inc. Warrenton, VA, USA).

## 3. Results and Discussion

### 3.1. Nutritional Composition of the Colada Beverage

Beverages based on liquid whey mixed with fruit juices are a great way to use a by- product of the cheese industry which is full of nutrients. Composition of the main ingredient, the whey, depends on the origin of the milk, type of cheese, and the cheese processing method [33]. Based on pH, whey may be classified as acidic whey (pH < 5) or sweet whey (pH 6–7); since acidic pH may cause protein sedimentation due to heat treatment, sweet whey is considered more suitable as the raw material for the whey-based beverages [15].

The chemical composition of the whey beverage is depicted in Table 1. The Ecuadorian technical standard NTE INEN 2609 establishes the requirements for whey-based beverages, i.e., those whose main ingredient is whey and are intended for direct consumption (FAO, 2022). The protein content found in the *Colada* (1.29%) significantly exceeds the minimum percentage established in Ecuadorian regulations of 0.4%, which reinforce its nutritional value. The result is within the range found in the bibliography for whey beverages, which may vary from 0.94% in beverages prepared with hydrolyzed collagen, milk powder and sucrose [34] to 1.21–1.61% in traditional whey blended with fruit juices [35].

The *Colada* beverage showed an ash content of around 0.5%; this value, that provides an estimate of the minerals present in the beverage, was slightly higher than the average content observed for whey (0.25–0.4%, [36]). Whey is recognized as a source of valuable minerals, especially sodium and potassium, followed by calcium and magnesium [37]. Due to the addition of barley, the *Colada* beverage also provided high amounts of phosphorus, which may reinforce the mineral contribution to the diet.

Analysis of the amino acid content revealed that the proteins of the *Colada* are rich in essential amino acids, with the exception of valine and methionine (tryptophan was not determined) (Table 2). The major amino acids were glutamic, aspartic, leucine, proline and lysine. Methionine and histidine were the limiting amino acids. The amino acid composition of the beverage is mainly determined by its principal component, sweet whey. The amino acid composition of whey depends on the milk origin and on the stage of lactation that influences on the β-lactoglobulin/α-lactoalbumin ratio [38].

### 3.2. Proteolysis and Peptide Release during In Vitro Digestion

Once the nutritional composition of the Colada had been established, the beverage was digested, following the INFOGEST protocol in order to study the DH and the changes in the protein profile. A negative control of the digestion was performed, with no enzymes to confirm there is no protein autolysis during the process (data not shown). The proteolysis started after the gastric phase with a 14 ± 2% DH and was much higher after the intestinal phase, with a 57.69 ± 3% DH measured by OPA. The protein profile after in vitro digestion was studied by SDS-PAGE (Figure 1) and exclusion molecular chromatography (SEC) (Figure 2). As expected, there was no protein degradation after the oral phase being the protein profile identical to the undigested *Colada* (lanes 2 and 3). Proteolysis started with the action of pepsin during the gastric digestion (lane 4). Most proteins of a molecular weight higher than 24 KDa were completely digested after 2 h. Two major protein bands at an apparent molecular weight around 21 and 14 KDa were observed, corresponding to β-lactoglobulin and α-lactoabumin. These two proteins have a globular structure which prevents the access of enzymes to potential cleavage sites and have been described as resistant to the action of the pepsin [39,40]. The great number of proteins of apparent molecular weight higher than 20 KDa, observed in the SDS-PAGE after the complete in vitro digestion, correspond to pancreatin background (lane 5). Most of the proteins are digested, and only small peptides are present after the digestive process.

In order to have a better picture of the peptide profile after digestion, SEC was used, and a control sample (digestion without sample) was ran and subtracted from the chromatogram of the completely digested sample to reduce the enzyme background observed in SDS-PAGE. The SEC chromatograms give valuable information on size distributions of proteins, large peptides, and smaller protein degradation products. The SEC profile of the *Colada* (Figure 2) indicated that the main difference in the oral phase (part A) compared to the gastric phase (part B) was observed in the range of 40–13 KDa (130–180 mL of elution volume). The two main peaks observed at 145 mL and 165 mL probably corresponded to β-lactoglobulin and α-lactoalbumin that were still undigested in the gastric phase, as it was observed in the SDS-PAGE. Although the digestion process starts in the gastric phase, the great majority of proteins were digested in the intestinal phase (part C). After this phase there was a great increase in the intensity in the peaks corresponding to peptides ≤6 KDa (240 mL of elution volume) and all the proteins with an apparent molecular weight ≥40 KDa were hydrolyzed into smaller peptides/free aa.

### 3.3. Antioxidant Properties

Currently, it is widely accepted that the effect of any dietary component is defined by the bioavailable amount rather than the dose ingested, and that bioaccessibility constitutes a prerequisite and the most influential factor of bioavailability [41]. Thus, determination of bioaccessibility in vitro reflects reasonably well bioavailability in vivo [42].

The digestive process had a general positive effect on the antioxidant properties of the *Colada*, represented by significant increases of around 2 fold on ABTS and FRAP assays after digestion, although a slight decline was observed in DPPH values (Table 3). Moreover, a strong increment of TPC was observed (>6 fold).

Both antioxidant properties and polyphenols post digestion were mainly located in the bioaccessible fraction (90–97% of ABTS, DPPH and FRAP; 80% of polyphenols, Figure 3), which included compounds released from the food matrix that are potentially absorbable from the lumen. In the residual fraction, usually discarded when studying bioavailability of bioactive compounds, there still remained low but appreciable amounts of antioxidant activity (10% of radical scavenging ability) and phenolic compounds (20%). These may exert antioxidant and anti-inflammatory local action [43] or be metabolized by the intestinal microbiota in potentially absorbable complexes [44].

Usually, antioxidant properties are measured in vitro by the ability to transfer hydrogen atoms or electrons to an oxidant, and ABTS, DPPH and FRAP methods are commonly applied with this aim. However, the use of cell model systems, such as the Caco-2 intestinal cell line, is recommendable to better assess the in vivo antioxidant potential of dietary compounds [37]. Preincubating Caco-2 cells for 2 h with the BF of the *Colada* beverage significantly reduced ROS generation in basal conditions (Figure 4A). In addition, a protective effect against induced oxidative stress (25% of decrease in ROS production) was observed (Figure 2B), supporting the antioxidant activity of the *Colada* digest at the cell level.

Our results showing the increasing release of bioactive antioxidant compounds during digestion agree with those previously observed in oils [21,27] and other foods [45], although disappointing data have also been observed in coffee extracts [46]. The antioxidant behavior of dietary components during digestion seems to be strongly dependent on the food matrix and the kind of antioxidant compounds; phytochemicals (carotenoids and polyphenols, among others) from fruit and vegetables are traditionally designed as the main contributors [47]. In the last years, whey-derived peptides with oxidation inhibitory activities have received special attention [37,48]. Although antioxidant properties of whey peptides and their hydrolysates have been deeply studied, little information exists on the effect of the digestion process and their interactions with other components when whey is included as a part of a complex food or beverage. Previous studies [48] demonstrated that antioxidant properties of isolated bovine whey proteins, tested by ABTS, FRAP and ORAC methods, significantly increase after simulated gastrointestinal digestion, α-lactalbumin having the highest antioxidant post-digestion activity. Therefore, the increasing radical scavenging ability (by ABTS) and ferric reducing power found in the present assay could be at least partly due to the digested whey proteins of the *Colada*, mainly α-lactoalbumin, as indicated by the results obtained by SDS-PAGE.

Interestingly, in Caco-2 cells, isolated digested whey proteins were unable to reduce induced ROS formation [8], although were effective in HT-29 cells and for stimulating antioxidant enzymes. Thus, in our study, other food components of the *Colada* digest could have contributed to the protecting ability against radical generation observed in Caco-2 cells.

It has been shown that combination of whey with additional antioxidant ingredients (vitamin B12, astaxanthin and plant extracts) improves the bioactivity of the formulated beverages [49], and that values of antioxidant activity in whey-based drinks added with fruit and vegetables are positively correlated with the polyphenol content [17]. Results of the present assay are comparable with those found for homemade beverages prepared with 50% whey and with fruits (pear, cherry and banana) and vegetables (carrot and parsley) added, in which DPPH and ORAC values were around 0.78 and 0.71 mM trolox Eq./kg, respectively [17]. Therefore, it seems clear that antioxidant properties of the intact *Colada* were due, in addition to the whey proteins, to antioxidant compounds, especially polyphenols, provided by barley and Maracuyá [18,19]. Moreover, polyphenol extracts of Maracuyá have shown preventive effects against induced barrier dysfunction in Caco-2 cells by increasing transepithelial electrical resistance [50], a reliable indicator of the normal and functional state of cell membranes which is linked to reducing level of oxidative stress and ROS accumulation [51].

However, these antioxidant properties may be altered during gastrointestinal transit. Interestingly, it has been reported that antioxidant activity after simulated digestion increases in whey beverages but decreases when drinks were added with additional antioxidant compounds [49]. It seems that adding carotenoids and polyphenols interact with the digestibility of whey proteins, reducing the release of antioxidant peptides and amino acids [18] which, in turn, decrease the antioxidant activity of the formulation [49]. This fact would explain the decreasing value of DPPH post digestion found in the *Colada* beverage. However, the possible negative influence of antioxidant phytochemicals seems to be counteracted with the high antioxidant potential of the whey proteins, since whey was the major component of the formulation, leading to strongly increased ABTS and FRAP values. On the other hand, discrepancies between different antioxidant assays (ABTS and FRAP) with DPPH have been previously observed [48].

The digestive process also produces structural transformations in food polyphenols, mainly due to changes of pH and interactions with other compounds, which may alter their bioaccessibility [52]. Anthocyanins, the major polyphenols of Maracuyá, are highly stable during simulated gastric digestion but are degraded to different catabolic products during the intestinal phase [53]. Therefore, the large increase found in TPC of the *Colada* BF could be due to the different chemical structures of the existing polyphenols and their biotransformation during digestive conditions. In addition, the fact that the Folin–Ciocalteu reagent, used for the determination of polyphenolic compounds, could react with amino acids from whey digest leading to overestimation of results [17], cannot be discarded.

### 3.4. The Lipid Fraction

The type of dietary fatty acids (FA) affects the physiological body response; saturated FA (SFA) have been related to adverse health effects, whereas unsaturated fatty acids, especially monounsaturated (MUFA) and n3 polyunsaturated (PUFA), are thought to be protective. They reduce the risk of cardiovascular and inflammatory diseases, although PUFA are more susceptible to being oxidized [54].

The FA profile and the related indexes of the *Colada* beverage before and after the in vitro digestion (BF) are shown in Table 4 and Figure 5, respectively.

Before digestion, the major FA of the *Colada* was palmitic acid (C16:0, 32%), followed by oleic (C18.1n9, 28%) and stearic acid (C18:0, 17%); this is in agreement with the main FA observed in whey [55] and in milk fat [56]. Total SFA represented the major fraction of the undigested sample (65%), whereas MUFA and PUFA contributed 31 and 4%, respectively. After digestion, significant changes in the FA profile of the beverage were observed: SFA were drastically reduced nearly to the half, whereas total MUFA and PUFA increased approximately 2- and 3-fold, respectively, compared with values pre-digestion.

Compositional studies of whey and whey beverages have been mainly focused on the protein components, and little attention has been given to the lipid fraction, probably due to its minority presence in the final product. Moreover, there is a lack of information concerning the FA profile of whey or whey-based drinks after the digestive process and the subsequent bioavailability. Santillo et al. [57] showed that the percentage distribution of FA in in vitro-digested milk from different species did not reflect the patterns of the corresponding milk sources, which is according with our results. This finding might be explained by different reasons. On the one hand, Ca may affect FA bioaccessibility since, at pH intestinal conditions, it reacts with saturated long chain FA (>14 C) and forms the corresponding insoluble soaps, which significantly reduces their availability [56]. Therefore, the high Ca level of the *Colada* could have induced the precipitation of myristic (C14:0), palmitic (C16:0), and stearic (C18:0) acid as non-soluble soaps in the RF, thus decreasing their levels in the soluble fraction.

On the other hand, the digestion may induce lipid oxidation, mainly due to acidic pH in the gastric fluid, the disintegration of the food matrix or the action of digestive enzymes [58]. Oxidation of PUFA led to the formation of lipid oxidation by-products and subsequent decrease of PUFA bioaccessibility, especially n3 [59]. However, after digestion of the *Colada*, higher levels of n6 and n3 in the BF were observed. This may be attributed to the high level of antioxidant compounds present in the formulation; specifically, it has been suggested that the addition of polyphenols during simulated digestion limits the lipid oxidation and promotes the bioavailability of non-oxidized PUFA [59]. Moreover, the food matrix also may impact on the release of FA during in vitro intestinal digestion. In this line, it has been shown that adding barley to a meal increases the release of long-chain PUFA and their percentage contribution to the total FA in the digest [60]. Therefore, the ingredients added to the whey beverage *Colada*, such as barley and Maracuyá polyphenols, could have contributed to increase the PUFA proportions in the BF after the in vitro digestion.

After digestion of *Colada*, strong increases in C18:1n9 and C18:2n6 were also observed. These FA are of interest in human nutrition, since oleic acid has noticeable health effects in preventing degenerative diseases, and linoleic acid an essential FA precursor of other important long chain FA [61]. Moreover, substitution of dietary saturated fat by oleic acid and/or PUFA produces cardiovascular benefits by reducing blood lipids [61].

Hence, the in vitro digestion of the *Colada* induces positive transformation in the composition and bioaccessibility of the FA, which could be at least partly due to the presence of antioxidant compounds provided by its different ingredients.

### 3.5. Peptide Identification with Antioxidant Potential Activity

Since antioxidant properties of the *Colada* could be also determined by the protein composition, we analyzed the peptides released after the in vitro digestion. Bioactive peptides are encrypted amino acid sequences inactive in the native protein that could be released through different processes including gastrointestinal digestion [62]. Ingested proteins from the *Colada* were hydrolyzed by proteinases present in the gastrointestinal tract that are used in the in vitro digestion process to produce peptides of various sizes, as shown in Figure 2. To identify peptides potentially responsible for the antioxidant activity observed in the digested sample, the bioaccesible fraction was analyzed by NanoLC-MS/MS. A total of 251 peptides of different lengths from 5 to 25 aminoacids were identified. The peptides released after the digestion of the *Colada* beverage not only originated from whey proteins but also from barley (Hordeon Vulgare) used in the formulation. Surprisingly, many of the peptides identified were fragments of β-casein normally not present in sweet whey. The presence of casein peptides in the digestion of the beverage made using sweet whey, suggests the leakage of casein into the whey fraction in the process of cheese making, a fact that has been previously described [8]. Peptides from κ-casein were also identified and the longer peptides were obtained from α-lactoalbumin and β-lactoglobulin. These last proteins are described as the least digestible proteins from whey. The sequences of the generated peptides were analyzed through BIOPEP a database of biologically active peptide sequences [31]. The peptides analyzed showed homology with several bioactive peptide sequences, whose activities include ACE inhibition, DDP-IV enzyme inhibition, antithrombotic, antibacterial and antioxidant effects. Here we wanted to highlight those peptides with antioxidant activity (Table 5).

Antioxidant peptides derived from milk proteins have been mostly associated with bovine casein [63,64]. The in vitro digestion of the *Colada* produced 11 peptides with sequences described as antioxidant activity from β-casein (Table 5). The antioxidant activity of casein derived peptides has been extensively studied and described [63]. One of the peptides identified, YPFPGPI, which shares sequence with other peptides found, has been recently described as a potent antioxidant [65]. Although antioxidant peptides derived from milk proteins have been mostly associated with bovine casein, hydrolysis of whey proteins may also result in the production of antioxidant peptides [63,64]. In our digested beverage we have identified 18 peptides from β-lactoglobulin, all with different biological activities but only 7 with potential antioxidant activity (Table 5). β-lactoglobulin is the major whey protein and biological active peptides derived from this protein is an area of intense research [66,67]. Four of the longer peptides identified (QTMKGLDIQKVAGTWYSLAMAASD, VAGTWYSLAMAASDISLLDAQSA, GLDIQKVAGTWYSLAMAASDISLL, DIQKVAGTWYSLAMAASDISLLDA) contain in their sequence previously described potent antioxidant peptides (WYSLAMAASDI, WYSLAMA, WYSLAM, WYSLA, WYSL, WYS, and WY). One of these peptides, WYSLAMAASDI, possessed higher radical scavenging activity than butylated hydroxyanisole (BHA), a potent antioxidant used in the food industry [68]. Only two of the sequenced peptides, GLDIQKVAGTWYSLAMAASDISLL and DIQKVAGTWYSLAMAASDISLLDA have a relatively high enough FRS score, predicted through AnOxprep-1.0, to be considered as potential antioxidant peptides. The presence of tyrosine and tryptophan could determine its potent activity, as these two amino acids have been reported to be the main contributors to the peroxyl radical-scavenging activity of food-derived peptides [69]. All of these peptides could still be considered as potential antioxidants since they could be further hydrolyzed by peptidases present in the intestinal brush border. The action of the enzymes present in the brush border is very intense and could release new peptides available for absorption with important biological activity [70,71].

The second most important milk protein, α-lactalbumin, is also known to be an important source of essential amino acids that present bioactive peptides encrypted in their sequence [8]. Ten peptides were identified from α-lactalbumin, all with previously described biological activities but none of them noted as antioxidant. Previous studies have shown that the antioxidant activity of α-lactalbumin increases after hydrolysis with alcalase and even more when this hydrolysate was subjected to a gastrointestinal digestion in vitro [72,73]. Very recently the sequence of these peptides has been reported [74]. Five of the peptides identified in our samples, LDDDL, LDDDLTDD, DDDLTDDI, LDDDLTDDI and DDDLTDDIM are fragments of the peptides previously identified as potent antioxidants (Table 6). The peptide VSLPE is a fragment of the antioxidant peptide VSLPEW sequenced earlier [75].

Despite the fact that α-lactoalbumin has been described as a potent antioxidant [12,69,75] only a few peptides have been described and none of them obtained after gastrointestinal digestion. Previous studies on peptides from this protein obtained by hydrolysis with Corolase PP^®^, a complex mixture of enzymes similar to pancreatin, proved its antioxidant activity but did not purify or identify any of the responsible peptides in the hydrolyzed fraction [69]. It is well known that the potential antioxidant activity of the peptides depends on the abundance of certain amino acids and the order in the sequence. Basic and acidic amino acid residues can act as hydrogen donors, and hydrophobic residues facilitate access to hydrophobic radical species. Hydrophobic amino acids such as Ala (A), Pro (P), Val (V), Ile (I), Leu (L), Phe (F), Tyr (Y) can contribute on antioxidant properties [12]. All the peptides found from α-lactalbumin have leucine present in its sequence, and could be in theory considered potential antioxidants, but the predictions using the Anoxprep-0.1 tool gives very low values for these peptides (Table 6). From our results, we can conclude that the peptides obtained after digestion of α-lactalbumin in the *Colada* need further investigation in order to establish their antioxidant potential.

Twenty-three peptides of barley proteins were also identified in our samples (Table 7).

None of these peptides were reported as antioxidant in the BIOPEP database, probably because it is a protein source not as well studied as milk. In order to establish the potential bioactivity of the sequenced peptides, these were analyzed using PeptideRanker. This bioactive peptide predictor provides a computational prediction, assigning scores between 0 and 1 to peptides based on the probability of being bioactive from the N-to-1 neural network (N1-NN). This is computed based on the peptide primary sequence trained from different bioactive peptide databases. Any peptide possessing a score above the 0.5 threshold is labeled as bioactive [76]. Sixteen of these peptides had a score above 0.5 and three of them, PQQPFP, PQQPPFG, QPQPFP, with a very high probability of being biologically active with a score above 0.9. Those peptides labelled as potentially bioactive were analyzed for their potential antioxidant activity using the Anoxprep-0.1 tool (Table 7). All of the peptides showed a moderate probability of being antioxidants, but this will require further investigation.

## 4. Conclusions

The present study represents an effort to create a whey-based functional food and is in accordance with the Sustainable Development Goals and the circular economy concept. Findings of the present study provide important insights into the potential antioxidant value of a sweet whey-based beverage, prepared by adding fruits (Maracuyá) and cereals (barley). Our results show that the digestive process stimulated the antioxidant properties of the beverage, and that the bioaccessible fraction obtained afterwards in vitro digestion was even able to prevent the induced ROS generation in intestinal cultured cells. Not only bioactive compounds resulting from the digestion of the sweet whey, but also from Maracuyá and barley, seem to contribute to the high antioxidant potential of the *Colada* beverage. A total of 251 peptides of 5–25 amino acids were identified after the digestion process; among them, 18 peptides originated from whey proteins (11 peptides from β-casein and 7 from β-lactoglobulin) and had a well-established antioxidant activity, but also 16 peptides from barley proteins were detected as potential antioxidants. We may conclude that the *Colada* beverage could be a valuable food with economic and nutritional advantages, since it may be a useful tool for profiting from a cheese by-product, many times discarded, causing contaminant problems and at the same time serving as a source of bioactive compounds with antioxidant properties.

## Figures and Tables

**Figure 1 antioxidants-11-01827-f001:**
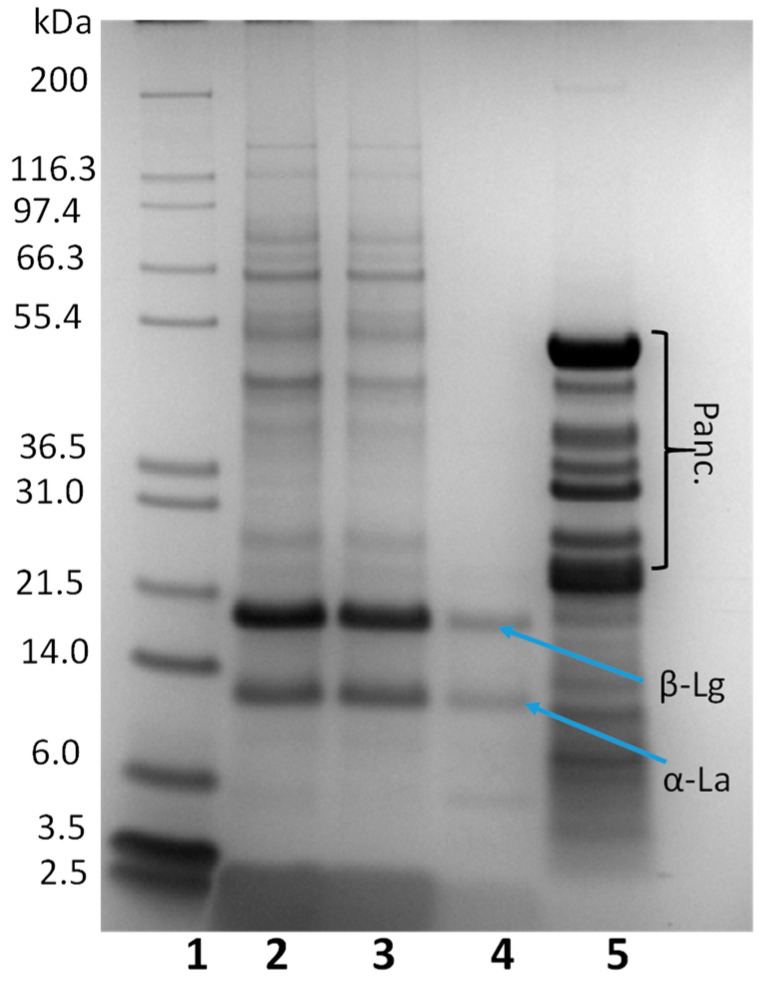
Protein profile of the whey-beverage (*Colada*) before and after in vitro digestion using an SDS-PAGE (4–12%) gel. Lane 1: molecular markers; Lane 2: *Colada* before digestion; Lane 3: oral phase; Lane 4: gastric phase; Lane 5: intestinal phase. Significant protein bands are denoted as follows: SA: serum albumin (66 kDa); β-Lg: β-lactoglobulin (18.3 kDa); α-La: α-lactalbumin (14.2 kDa); Panc. refers to proteins present in Pancreatin.

**Figure 2 antioxidants-11-01827-f002:**
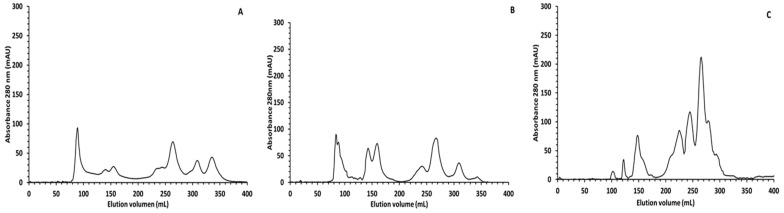
Size exclusion chromatography profile of protein and peptides after different phases of digestion of the whey-beverage (*Colada*). (**A**) Oral phase, (**B**) gastric phase, (**C**) intestinal phase (complete digestion).

**Figure 3 antioxidants-11-01827-f003:**
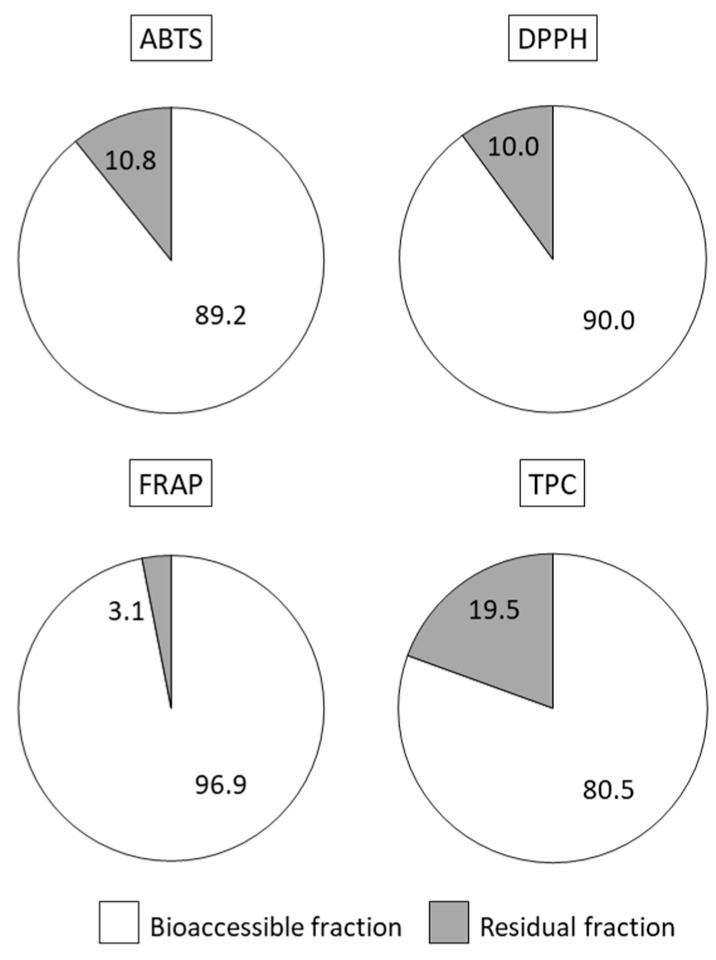
Contribution of the bioaccessible and residual fractions (%) to the total ABTS, DPPH, FRAP and total polyphenols (TPC) recovered after in vitro digestion of the whey beverage (*Colada*).

**Figure 4 antioxidants-11-01827-f004:**
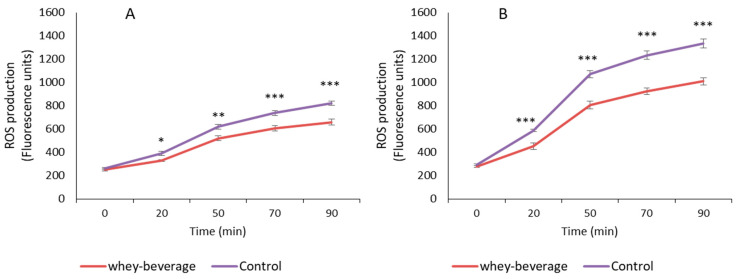
ROS generation expressed as fluorescence units during 90 min (mean ± SE) in Caco-2 cells pre-incubated with BF of the whey beverage compared with control cells incubated with culture medium. (**A**) Basal conditions. (**B**), oxidative stress induced with t-BOOH 10 mM. *, *p* < 0.05; **, *p* < 0.01; ***, *p* < 0.001.

**Figure 5 antioxidants-11-01827-f005:**
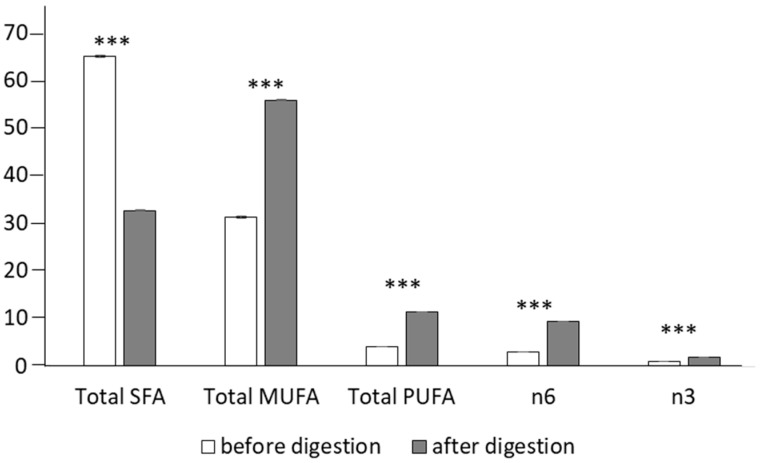
Fatty acid indexes before and after the in vitro digestion of the whey-beverage; SFA: saturated fatty acids; MUFA: monounsaturated fatty acids; PUFA: polyunsaturated fatty acids. Asteriscs means significant differences (one-way ANOVA and LSD test, *** *p* < 0.001).

**Table 1 antioxidants-11-01827-t001:** Composition and nutritional analysis of the whey-beverage (*Colada*).

**Ingredients (%)**	
Sweet whey	70
Maracuyá juice	15
Barley	10
Saccharose	4
Aromatic spices ^1^	1
**Chemical analysis ^2^**	
Moisture (%)	81.6 ± 0.001
Ash (%)	0.17 ± 0.001
Fat (%)	0.87 ± 0.08
Protein (%)	1.09 ± 0.05
Energy (Kcal/100 g)	81.9 ± 0.11
Calcium (mg/100 g)	29.9 ± 0.10
Phosphorus (mg/100 g)	39.6 ± 0.23
Magnesium (mg/100 g)	8.82 ± 0.07
Sodium (mg/100 g)	34.2 ± 0.20
Potassium (mg/100 g)	156 ± 1.20

^1^ Mix of equal parts sweet pepper, cloves and cinnamon. ^2^ Values are mean ± SE (n = 4), expressed on a fresh matter basis.

**Table 2 antioxidants-11-01827-t002:** Amino acid composition of the whey-beverage (*Colada*).

Amino Acid	mg/g *Colada* ^1^
Alanine	0.544 ± 0.057
Arginine	0.461 ± 0.147
Asx	1.190 ± 0.19
Cysteine	0.309 ± 0.126
Glx	2.496 ± 0.303
Glycine	0.344 ± 0.036
Histidine	0.286 ± 0.061
Isoleucine	0.623 ± 0.065
Leucine	1.089 ± 0.091
Lysine	0.903 ± 0.113
Metionine	0.230 ± 0.070
Phenilalanine	0.603 ± 0.107
Proline	0.993 ± 0.103
Serine	0.599 ± 0.066
Threonine	0.682 ± 0.062
Tyrosine	0.367 ± 0.064
Valine	0.659 ± 0.123

^1^ Values are mean ± SE (n = 3), expressed on a fresh matter basis. Asx: aspartic acid (Asp) and asparagine (Asn); Glx: glutamic acid (Glu) and glutamine (Gln). Cystein is measured as cysteic acid and methionine as methyl sulfone. Tryptophan was not determined due to its degradation after acid hydrolysis.

**Table 3 antioxidants-11-01827-t003:** Antioxidant activity by ABTS (2,2′-Azino-bis(3-ethylbenzothiazoline-6-sulfonic acid)), DPPH (2,2 diphenyl-1-picrylhydrazyl) and FRAP (ferric reducing antioxidant power), and total phenolic content (TPC) of the whey-beverage (*Colada*) before and after the in vitro digestion.

	Before Digestion	After Digestion		Fold Increase
		Bioaccessible Fraction	Residual Fraction	
ABTS (mM trolox Eq./kg)	0.76 ± 0.001 ^b^	1.21 ± 0.04 ^c^	0.15 ± 0.001 ^a^	1.79
DPPH (mM trolox Eq./kg)	0.51 ± 0.01 ^c^	0.41 ± 0.001 ^b^	0.05 ± 0.004 ^a^	0.90
FRAP (mM trolox Eq./kg)	0.74 ± 0.01 ^b^	1.66 ± 0.07 ^c^	0.05 ± 0.01 ^a^	2.31
TPC (mM galic acid Eq./kg)	0.46 ± 0.02 ^a^	2.42 ± 0.12 ^b^	0.58 ± 0.04 ^a^	6.52

Values are mean ± SE (n = 4). Means values in each row with different letters are significantly different (one-way ANOVA and LSD test, *p* < 0.05).

**Table 4 antioxidants-11-01827-t004:** Fatty acid profile (%) of the whey-beverage (*Colada*) before and after the in vitro digestion.

Fatty Acid	Before Digestion	After Digestion (Bioaccessible Fraction)	Fold Increase
C8:0	0.02 ± 0.001 ^a^	0.07 ± 0.02 ^b^	3.71
C10:0	0.68 ± 0.007 ^a^	2.14 ± 0.05 ^b^	3.15
C12:0	1.81 ± 0.03 ^a^	1.95 ± 0.02 ^b^	1.07
C14:0	9.82 ± 0.11 ^b^	5.64 ± 0.11 ^a^	0.58
C14:1	0.78 ± 0.004 ^a^	1.22 ± 0.003 ^b^	1.57
C16:0	31.6 ± 0.12 ^b^	15.3 ± 0.03 ^a^	0.48
C16:1	1.83 ± 0.03 ^a^	3.39 ± 0.06 ^b^	1.85
C17:0	1.04 ± 0.004 ^b^	0.28 ± 0.008 ^a^	0.27
C17:1	0.59 ± 0.05 ^a^	2.50 ± 0.03 ^b^	4.25
C18:0	17.3 ± 0.05 ^b^	7.23 ± 0.03 ^a^	0.42
C18:1 n9	27.6 ± 0.24 ^a^	45.4 ± 0.40 ^b^	1.68
C18:1 n7	0.50 ± 0.003 ^a^	0.96 ± 0.01 ^b^	1.93
C18:2 n6	2.97 ± 0.04 ^a^	8.91 ± 0.009 ^b^	3.00
C18:3 n6	0.02± 0.003 ^a^	0.04 ± 0.009 ^b^	2.60
C18:3 n3	0.84 ± 0.003 ^a^	1.57 ± 0.005 ^b^	1.88
C20:0	0.30 ± 0.004 ^b^	0.10 ± 0.001 ^a^	0.32
C20:3 n6	0.04 ± 0.001	0.08 ± 0.023	2.25
C20:3 n3	0.08 ± 0.02	0.11 ± 0.004	1.47
C20:4 n6	0.06 ± 0.001 ^a^	0.33 ± 0.02 ^b^	5.46
C20:5 n3	0.05 ± 0.005 ^a^	0.08 ± 0.006 ^b^	1.58
C21:0	1.51 ± 0.007 ^a^	2.42 ± 0.01 ^b^	1.60
C22:4 n6	0.008 ± 0.002 ^a^	0.04 ± 0.005 ^b^	4.67
C22:5 n3	0.09 ± 0.001 ^a^	0.17 ± 0.005 ^b^	1.99
C22:6 n3	0.01 ± 0.001	0.02 ± 0.001	1.73

Values are mean ± SE (n = 3). Means values in each row with different letters are significantly different (one-way ANOVA and LSD test, *p* < 0.05).

**Table 5 antioxidants-11-01827-t005:** Peptides with antioxidant activity identified in the bioaccessible fraction of the digested whey beverage (*Colada*).

Parental Protein	Identified Peptide Sequence	BIOPEP Identified Peptide	Peptide ID	Location
**β-Casein**	LPLPL	LPL	10000	[1,2,3,4,5]
	YQEPV	YQEP	7878	[1,2,3,4]
	YPFPGPI	YPFPGPI	10050	[1,2,3,4,5,6,7]
	ELNVPGEIa	EL	7888	[1,2]
	VYPFPGPI	VY	8224	[1,2]
		YPFPGPI	10050	[2,3,4,5,6,7,8]
	VYPFPGPIP	VY	8224	[1,2]
		PFPGPI	10050	[2,3,4,5,6,7,8]
		PFPGPIP	10062	[2,3,4,5,6,7,8,9]
	LVYPFPGPI	VY	8224	[2,3]
		YPFPGPI	10050	[3,4,5,6,7,8,9]
	SLVYPFPGPI	VY	8224	[3,4]
		YPFPGPI	10050	[4,5,6,7,8,9,10]
	PPTVMFPPQSVLSLSQSKV	SLV	9879	[10,11,12]
	QSLVYPFPGPIHNSLPQNIPPL	VY	8224	[4,5]
		YPFPGPI	10050	[5,6,7,8,9,10,11]
**β-lactoglobulin**	QTMKGLDIQKVAGTWYSLAMAASD	WY	7898	[15,16]
		WYS	7900	[15,16,17]
		WYSL	7901	[15,16,17,18]
		WYSLA	7902	[15,16,17,18,19]
		WYSLAM	7903	[15,16,17,18,19,20]
		WYSLAMA	7905	[15,16,17,18,19,20,21]
		TW	8459	[14,15]
		GTW	9165	[13,14,15]
		VAGTWY	9435	[11,12,13,14,15,16]
	DIQKVAGTWYSLAMAASDISLLDA	WYSLAMAASDI	7891	[9,10,11,12,13,14,15,16,17,18,19]
		WY	7898	[9,10]
		WYS	7900	[9,10,11]
		WYSL	7901	[9,10,11,12]
		WYSLA	7902	[9,10,11,12,13]
		WYSLAM	7903	[9,10,11,12,13,14]
		WYSLAMA	7905	[9,10,11,12,13,14,15]
		TW	8459	[8,9]
		GTW	9165	[7,8,9]
		VAGTWY	9435	[5,6,7,8,9,10]
	VAGTWYSLAMAASDISLLDAQSA	WYSLAMAASDI	7891	[11,12,13,14,15,16,17,18,19,20,21]
		WY	7898	[11,12]
		WYS	7900	[11,12,13]
		WYSL	7901	[11,12,13,14]
		WYSLA	7902	[11,12,13,14,15]
		WYSLAM	7903	[11,12,13,14,15,16]
		WYSLAMA	7905	[11,12,13,14,15,16,17]
		TW	8459	[10,11]
		GTW	9165	[9,10,11]
		VAGTWY	9435	[7,8,9,10,11,12]
	TMKGLDIQKVAGTW	TW	8459	[13,14]
		GTW	9165	[12,13,14]
	LDTDYKKYLLFCMENSAE	TDY	8445	[3,4,5]
		FC	9342	[11,12]
		YLL	9349	[8,9,10]
		LFC	9352	[10,11,12]
		FCM	9353	[11,12,13]
		CME	9354	[12,13,14]
		DYK	9364	[4,5,6]
		KKY	9365	[6,7,8]
		KYL	9366	[7,8,9]

**Table 6 antioxidants-11-01827-t006:** Peptides identified from alpha-lactoalbumin.

Peptide Identified from Alpha-Lactoalbumin	Free Radical Scavenging Score	Chelating Score	Peptide Reported in Literature
CTTFHTSGYDTQAIVQNNDS	0.357	0.211	YDTQA ^1^
TTFHTSGYDTQAIVQNNDSTEYG	0.325	0.236	
GVSLP	0.376	0.237	
LDKVG	0.315	0.210	
VSLPE	0.402	0.267	VSLPEW ^1^
LDDDL	0.320	0.237	DKFLDDDLTDDIM, KFLDDDLTDDIM, IWCKDDQNPH ^2^
LDDDLTDD	0.276	0.256	
DDDLTDDI	0.283	0.268	
LDDDLTDDI	0.274	0.247	
DDDLTDDIM	0.273	0.270	

References in the literature ^1^: Sadat et al., [75]; ^2^: Báez et al., [74].

**Table 7 antioxidants-11-01827-t007:** In silico characterization of peptides from barley identified in the bioaccessible fraction of digested whey beverage (*Colada*). Antioxidant activity prediction by AnOxPrep.

Parental Protein	Peptide Sequence	Peptide Ranker Score	Free Radical Scavenging Score	Chelating Score
B1-hordein	PQQPFPPQQPF	0.88	0.52	0.35
B1-hordein	KPFPQQP	0.62	0.42	0.30
B1-hordein	PQQPFPPQQP	0.71	0.52	0.34
B1-hordein	PQQPPFG	0.92	0.51	0.27
B1-hordein, B3-hordein	LPQIPE	0.31		
B1-hordein, B3-hordein	QLPQIPEQ	0.22		
B1-hordein, B3-hordein	LPQIPEQ	0.21		
B1-hordein, B3-hordein, C-hordein	PQQPIPQ	0.51	0.46	0.27
B1-hordein, B3-hordein, C-hordein	PQQPIP	0.66	0.46	0.27
B1-hordein, C-hordein, Gamma-hordein	PQQPFPQ	0.75	0.48	0.28
B1-hordein, C-hordein, Gamma-hordein	PQQPFP	0.9	0.48	0.26
B3-hordein	QPFPQQP	0.69	0.45	0.32
B3-hordein	QVQIP	0.17		
C-hordein	PQQPLP	0.73	0.46	0.27
C-hordein	PQQPLPQ	0.52		
C-hordein	SQQPIPQ	0.46		
C-hordein	QPLPQPQ	0.48		
C-hordein, B3-hordein	QPQPFP	0.92	0.48	0.28
C-hordein, B3-hordein	QPQPFPQ	0.8	0.48	0.29
Gamma-hordein, C-hordein	QPFPQP	0.87	0.47	0.31
Gamma-hordein, C-hordein	PQQPFPQQPQQPFP	0.66	0.51	0.33
Gamma-hordein	HQFPQPT	0.58	0.47	0.29
Gamma-hordein	FNPSG	0.75	0.41	0.28

## Data Availability

Data is contained within the article or Appendix A.

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
