# Peer review of "Antioxidant Potential of the Sweet Whey-Based Beverage Colada after the Digestive Process and Relationships with the Lipid and Protein Fractions"

_antioxidants, 2022, doi:10.3390/antiox11091827_

Round 1

Reviewer 1 Report

Antioxidants. August 2022 review

Manuscript number Antioxidants-1880194

Title: Antioxidant potential of the sweet whey-based beverage Colada 2 after the digestive process and relationships with the lipid and protein fractions

This manuscript reports on the properties of Colada, a whey-based beverage combined with maracuya and barley. The authors combined different ratios of maracuya juice to make a sensorial analysis to select the most accepted whey-drink based on taste. The selected whey beverage was analysed organoleptically in terms of its attributes with a panel of 80 untrained consumers. The authors also studied the nutritional composition, in vitro digestibility (DH, gel filtration analysis, SDS-PAGE), total phenolic content, amino acid analysis, antioxidant activity (DPPH, ABTS and FRAP assays and ROS in cell culture) and fatty acid analysis. Moreover they also carried out antioxidant peptides identification using LC-MS/MS and different databases. The obtained results indicated that the digestive process increased the antioxidant activity, polyphenols, peptides and fatty acids. Authors identified several peptides of barley and whey with potential antioxidant activity.

Overall, the manuscript provided some interesting new information and it is clear and well written. However, a minor clarification should be included in Table 2 in the amino acid composition. During acid hydrolysis of the proteins to determine the amino acids, asparagine and glutamine are converted in aspartic acid and glutamic acid, respectively. Therefore, in the literature these amino acids are generally expressed as asparagine + aspartic acid (Asx) and glutamine + glutamic acid (Glx). So this should be indicated in Table 2. It is also recommended to repeat in this table that Trp was not determined since it is destroy during acid hydrolysis.

Other specific comments are as follows:

Tables 2, 6 and 7. The “,” should be changed to “.” To indicate the decimals.

Reviewer 2 Report

This is a detailed and full experimental study of a whey-based drink which uses a wide range of techniques to assess the chemical changes induced by an artificial. digestion method. The focus of the study is the potential for antioxidant activity.

The novelty of the study has not been demonstrated and although my preliminary search showed no previous similar studies,  it is for the authors to demonstrate this in a clear way.

It is also not clear how significant the results of this study would be. To show that antioxidant potential is retained and also increased a little after digestion is of course important but what is the real advance here? is it likely to lead to using whey in this drink? 

Throughout the study, the authors refer to the antioxidant potentials of proteins and polyphenols. Almost every organic and biochemical compound will react with free radicals and therefore be an antioxidant. The reader needs to know then what is a good antioxidant and what is not. The ABTS, FRAP, DPPH values require some context. What are the maximum values ( shown by a good antioxidant) and what are the minimum values normally seen for food components? 

Furthermore, polyphenols are usually seen as good antioxidants but some have SIRT-like activity. It would be useful to make reference to this difference for balance and recognition of an emerging field of research.

Author Response

-Reviewer 2

Comments and Suggestions for Authors

This is a detailed and full experimental study of a whey-based drink which uses a wide range of techniques to assess the chemical changes induced by an artificial digestion method. The focus of the study is the potential for antioxidant activity.

The novelty of the study has not been demonstrated and although my preliminary search showed no previous similar studies, it is for the authors to demonstrate this in a clear way.

It is also not clear how significant the results of this study would be. To show that antioxidant potential is retained and also increased a little after digestion is of course important but what is the real advance here? is it likely to lead to using whey in this drink?

Thank you for your constructive comments and suggestions. We agree that the novelty of the study, as well as the scientific and social significance that results could have, should be clearly remarked. 
As we have exposed in the Introduction, about 50 % of whey is currently wasted as a byproduct of the cheese industry (Pires et al., 2021, doi: 10.3390/foods10051067). This fact, in addition to causing serious environmental hazards, implies the loss of a food ingredient with very high nutritional and healthy possibilities for consumers, which is especially important in developing countries. This work has been carried out in collaboration with one of those countries, Ecuador, in which the rates of malnutrition (especially among the child population) are striking.  In this sense, the present study represents an effort to create a whey-based functional food trying to profit nutritional and economic advantages.  The work is aimed to increase sustainability at the agri-food level and is in accordance with both the Sustainable Development Goals and the circular economy concept (https://www.un.org/sustainabledevelopment/sustainable-development-goals/ and https://ec.europa.eu/commission/presscorner/detail/en/IP_19_2391), as it is based in valorizing the resources obtained during the food production processes.   We have shown that replacing water by whey in a traditional beverage results in a healthy food, which antioxidant potential increases during the simulated digestion process. Although some studies have assessed the antioxidant properties of whey beverages (such as Purkiewicz and Pietrzak-Fiecko, 2021, doi.org/10.3390/molecules26113126), there was no information about changes occurred during the digestive processes. Another important novelty of our research, comparing with similar studies, is using the complete whey in a real beverage easy to prepare with hardly any cost, unlike other studies that have used isolated whey components (mainly proteins) which requires previous expensive technological preparation (encapsulation and others). In addition, we also show the importance of the food matrix in the final qualities of the digested product, as antioxidant properties of the whey beverage may be partly attributed to the fruit and cereals added. 
We have included some of these comments in the new version of the manuscript.   

Throughout the study, the authors refer to the antioxidant potentials of proteins and polyphenols. Almost every organic and biochemical compound will react with free radicals and therefore be an antioxidant. The reader needs to know then what is a good antioxidant and what is not. The ABTS, FRAP, DPPH values require some context. What are the maximum values (shown by a good antioxidant) and what are the minimum values normally seen for food components?

Furthermore, polyphenols are usually seen as good antioxidants but some have SIRT-like activity. It would be useful to make reference to this difference for balance and recognition of an emerging field of research.                                   

We thank the considerations of the referee regarding antioxidant compounds (also exposed in our recent paper “To Be or Not to Be An Antioxidant?, That Is the Question”, Palma and Seiquer, doi:10.3390/antiox9121234).

Of the in vitro methods for analyzing antioxidant activity in foods, ABTS, DPPH and FRAP are among the more used and cited in the literature, including research regarding antioxidant properties of whey beverages or whey proteins (cited in our paper, such as Arranz et al. 2019 [49], Corrochano et al., 2019 [5] or Purkiewicz and Pietrzak, 2021 [17]). However, the diverse form of expressing the results (in Trolox equivalents per unit of weight or of volume, on fresh or dry matter basis, per serving, as a percentage of inhibition compared with a control or calculating the IC50, which is the concentration of the sample required to inhibit 50% of radical … ) makes comparisons among authors enormously difficult. This is one of the reasons why there are no maximum or minimum values for these tests established in the literature, and we think that comparisons with values of other foods could confound to the reader. Papers showing antioxidant activity measured by the same methods in other food matrix have been included in our paper (references 21, 27 and 45), and the reader may compare the related results.

On the other hand, the aim of the study is to determine changes in antioxidant activity during the digestion process of the Colada beverage rather that compare with other foods and, for such purpose, values before and after the in vitro digestion are showed.

Reviewer 3 Report

The study is interesting but there are a lot of issues in the present version. Some of the analyses need more clarifications and all chromatograms from the GC and HPLC need to be included. The Mechanisms for the observed findings need to be discussed. Especially the increase of compounds as a result of digestion.

I included a large number of comments in the attached PDF for your attention.

Round 2

Reviewer 3 Report

Please in the attachment
